# The Role of the MAPK Signaling Pathway in Cardiovascular Disease: Pathophysiological Mechanisms and Clinical Therapy

**DOI:** 10.3390/ijms26062667

**Published:** 2025-03-16

**Authors:** Xueyang Wang, Ruiqi Liu, Dan Liu

**Affiliations:** 1Queen Mary College, Nanchang University, Nanchang 330001, China; xueyang.wang@se21.qmul.ac.uk (X.W.); ruiqi.liu@se22.qmul.ac.uk (R.L.); 2School of Pharmacy, Jiangxi Medical College, Nanchang University, Nanchang 330006, China

**Keywords:** cardiovascular disease, MAPK signaling pathway

## Abstract

Cardiovascular disease (CVD) is a serious global health issue with high mortality rates worldwide. Despite the numerous advancements in the study of CVD pathogenesis in recent years, further summarization and elaboration of specific molecular pathways are required. An extensive body of research has been conducted to elucidate the association between the MAPK signaling pathway, which is present in all eukaryotic organisms, and the pathogenesis of cardiovascular disease. This review aims to provide a comprehensive summary of the research conducted on MAPK and CVD over the past five years. The primary focus is on four specific diseases: heart failure, atherosclerosis, myocardial ischemia–reperfusion injury, and cardiac hypertrophy. The review will also address the pathophysiological mechanisms of MAPK in cardiovascular diseases, with the objective of proposing novel clinical treatment strategies for CVD.

## 1. Introduction

Cardiovascular diseases, including atherosclerosis and myocardial disease, constitute a major global health burden, contributing substantially to morbidity and mortality [1,2]. The Global Burden of Disease (GBD) study indicates that ischemic heart disease has remained the leading cause of mortality, exhibiting the highest age-standardized mortality rate from 1990 to 2021 [3]. Despite significant advancements in cardiovascular disease research over the past decade, a critical gap remains in accurately elucidating its pathogenesis [4]. It is therefore crucial to investigate the potential molecular signaling pathways that contribute to the etiology of cardiovascular disease.

Signaling pathways serve as fundamental mechanisms for transmitting external signals to the genome within the nucleus. External stimuli, including growth factors, cytokines, hormones, and cellular stress responses, elicit a chain reaction involving numerous proteins, including receptor proteins, protein kinases, and transcription factors [5]. The etiology of cardiovascular disease is dependent upon a number of signaling pathways, including the JAK-STAT pathway [6], the non-canonical WNT signaling pathway [4], the Nrf2/HO-1 signaling pathway [7], and the MAPK signaling pathway [8]. Among these, the mitogen-activated protein kinase (MAPK) signaling pathway is highly conserved and has been shown to regulate various biological processes, including metabolic reprogramming, cell proliferation, survival, and differentiation [9]. Extensive studies have provided substantial evidence linking the MAPK pathway to cardiovascular health and disease pathogenesis [10,11,12]. This provides us with potential therapeutic targets and modalities. Thus, a comprehensive understanding of the MAPK signaling pathway is crucial for developing effective therapeutic strategies for various cardiovascular diseases.

In this review, we explore the research conducted in recent years (primarily over the past five years) on the relationship between the MAPK signaling pathway and cardiovascular disease. First, we present an overview of the MAPK signaling pathway, followed by an analysis of its role in cardiovascular disease pathogenesis, with a particular focus on myocardial ischemia–reperfusion injury, heart failure, atherosclerosis, and cardiac hypertrophy. Finally, we discuss the potential clinical applications of targeting the MAPK signaling pathway in the treatment of cardiovascular disease.

## 2. Basic Information of the MAPK Signaling Pathway

Mitogen-activated protein kinases (MAPKs) are serine-threonine protein kinases that become activated in response to various stimuli, including cytokines, growth factors, neurotransmitters, hormones, cellular stress, and cell adhesion [13]. The MAPK family comprises three major subgroups: extracellular signal-regulated kinases (ERKs), p38, and c-Jun N-terminal kinases (JNKs), each representing a distinct signaling cascade [14]. The MAPK signaling cascade is among the most evolutionarily conserved pathways, characterized by a sequential phosphorylation process that ultimately triggers specific cellular responses upon activation [15]. The cascade is known to initiate with the activation of MAPK kinase kinase (MAPKKK), which subsequently phosphorylates and activates MAPK kinase (MAPKK). This process culminates in the activation of MAPK [15,16]. All eukaryotic cells possess multiple MAPK pathways that function in a coordinated manner to regulate diverse cellular processes, including gene expression, mitosis, metabolism, motility, survival, apoptosis, and differentiation [17]. The classification of MAPK can be further refined according to different structural and regulatory features, with the distinction being drawn between classical and atypical MAPK [18]. The process of signal transduction through MAPKK and MAPK activation following MAPKKK activation constitutes a classical MAPK signaling pathway, which is associated with the ERK1/2, ERK5, p38, and JNK1/2/3 signaling cascades. In contrast, other signaling processes are classified as atypical MAPK signaling pathways, which are regulated by ERK3/ERK4, ERK7/ERK8, and NLK signaling cascades [5].

The ERK pathway is among the most extensively studied signal transduction pathways in cell biology [19]. Its initial identification in the early 1980s was due to the ability to phosphorylate microtubule-associated protein-2 (MAP-2) in 3T3-L1 adipocytes in response to insulin stimulation [20]. Recently, Feng et al. reported that Versican-mediated activation of integrin β1 triggered the ERK1/2 pathway, promoting cardiomyocyte proliferation and cardiac repair [21]. Bassat et al. discovered that, in vitro, the activation of ERK-mediated signaling by recombinant Agrin promoted the division of cardiomyocytes derived from both mouse and human induced pluripotent stem cells, and that this process acted on cardiac repair [22]. These findings suggest a pathophysiological role for ERK in the heart.

The p38 MAPK signaling pathway is a complex system that enables cells to interpret a broad array of external signals and respond accordingly by generating a multitude of different biological effects [23]. P38 is a molecule that has been shown to bind to pyridylimidazole, thereby inhibiting the production of pro-inflammatory cytokines [24]. The p38 family can be categorized into two distinct subsets. The first subset consists of p38α and p38β, while the second subset comprises p38γ and p38δ [25]. There has been substantial research that unequivocally demonstrates the pathophysiological role of the p38 MAPK signaling pathway in cardiac health and disease [25]. Recently, as demonstrated by Kang et al., Lphn2-mediated activation of the p38-MAPK pathway exerts a regulatory effect on the connectivity of both mitochondria and cells in cardiomyocytes, thus contributing to the maintenance of cardiac integrity [26]. Collectively, these findings suggest a pathophysiological role for p38 in the cardiovascular system.

Approximately a decade after the discovery of ERK, JNK was identified as the second major subfamily of MAPKs [19]. This subfamily is further categorized into JNK1, JNK2, and JNK3, with each of these being encoded by a distinct gene [27]. As a stress-responsive signaling pathway, JNK plays a dual role in cardiomyocytes, exerting both protective and pathological effects [19]. Furthermore, a substantial body of research has demonstrated that the MAPK-ERK1/2 and JNK1/2 pathways are activated during heart failure by various factors, including growth factors, advanced glycation end products (AGEs), and angiotensin II (Ang II). This activation initiates a sequence of events that leads to cardiac fibroblast proliferation and collagen maturation [28].

In summary, current studies indicate that the MAPK signaling pathway is a crucial regulatory mechanism in eukaryotic organisms and plays a pivotal role in cardiovascular health and disease.

## 3. MAPK Signaling Pathway and Heart Failure

Heart failure (HF) is characterized by the heart’s impaired ability to pump blood efficiently, primarily due to weakened contractility or reduced elasticity of the cardiac muscle. This dysfunction manifests through various symptoms, including fatigue, dyspnea, weight gain, and edema in the extremities or abdomen [29]. HF affects approximately 1–2% of the adult population, making it a complex systemic disorder. The activation of various neurohormonal pathways, alongside structural changes at both macroscopic and microscopic levels—including alterations in cardiac dimensions, mass, and configuration—contributes to the progressive decline of cardiac function. In response to these changes, compensatory mechanisms involving humoral and molecular processes are activated to sustain cardiac homeostasis within a specific physiological range [30].

The MAPK signaling pathway is recognized as a crucial regulatory component in this process. Research indicates a correlation between HF of various etiologies and MAPK signaling [31,32,33,34]. A multi-omics study by Ouwerkerk et al. identified 177 enriched pathways associated with HF progression, with the MAPK/Akt pathway ranking among the most significantly enriched signaling networks. Furthermore, pathway interaction analysis identified significant associations between MAPK and key molecular complexes, such as GDF15-ERBB2 and VEGFR2-S1PR1-ERK1/2-PKC-alpha [35]. Moreover, an integrated analysis by Zhou et al. identified multiple biomarker modules strongly associated with dilated cardiomyopathy-induced HF, primarily enriched in the p38/MAPK and JAK-STAT pathways [32]. Zhang et al. demonstrated that pharmacological inhibition of the ERK/MAPK and TGF-β pathways significantly alleviated chronic HF in mice [36].

The MAPK signaling pathway has been implicated in the pathogenesis of HF through multiple pathophysiological mechanisms. In HF patients, myocardial fibrosis is a fundamental pathological mechanism driving myocardial remodeling. The MAPK signaling pathway, in particular, has emerged as a prominent avenue of investigation in this field [37]. Evidence suggests that overactivation of the ERK1/2 and JNK1/2 signaling pathways within the MAPK family synergistically promotes myocardial fibroblast proliferation and accelerates myocardial fibrosis progression [28]. These findings suggest that the MAPK signaling pathway may contribute to HF pathogenesis by modulating myocardial fibrosis. In vivo and in vitro studies indicate that inhibiting the MAPK signaling pathway, including p38, ERK1/2, and JNK, can mitigate ROS-induced tissue damage, including cardiac injury, and reduce oxidative stress. Consequently, this can ameliorate and prevent diabetes-related myocardial fibrosis [38]. GSK-3α-mediated activation of the RAF-MEK-ERK MAPK pathway in mouse cardiac fibroblasts has been shown to enhance the production of fibrotic proteins, such as IL-11 and collagen-1, promote fibroblast-to-myofibroblast transformation, and exacerbate injury-induced cardiac remodeling and dysfunction, thereby accelerating HF progression [39]. In conclusion, activation of the MAPK pathway may exacerbate HF progression by promoting myocardial fibrosis, which can be partially mitigated by MAPK inhibition.

Moreover, cardiac hypertrophy is a major predisposing factor for HF and can ultimately lead to its progression [40]. Zhang et al. demonstrated that Shexiang Tongxin Dropping Pill (STDP) inhibits the ERK/MAPK signaling cascade, thereby mitigating myocardial fibrosis and cardiomyocyte hypertrophy and subsequently preventing the progression of chronic HF in mice [36]. Similarly, Zhang et al. found that attenuating the MEK/ERK MAPK signaling pathway and downregulating *HOXA5* expression in a mouse model effectively prevented pathological cardiac hypertrophy and HF [41]. In an in vivo HF model, Peng et al. demonstrated that reducing mitochondrial ROS levels and inhibiting the p38 MAPK pathway alleviated myocardial hypertrophy and suppressed myocardial fibrosis, thereby mitigating HF progression [42].

Beyond its established role in cardiac hypertrophy and myocardial fibrosis, recent studies have uncovered additional pathophysiological mechanisms by which MAPK signaling contributes to HF progression. Xu et al. demonstrated that modulation of the intestinal microbiota composition, which in turn regulates key factors in the p38 MAPK signaling pathway, effectively delays ventricular remodeling in cardiomyocytes and enhances cardiac function in a rat model of chronic HF [43]. Moreover, Garlapati et al. identified overexpression of the MAPK pathway in human ischemic HF through quantitative phosphorylation proteomics of cardiac tissue. Furthermore, they demonstrated that inhibiting ERK1/2 activation improved IHF progression in a preclinical mouse model of myocardial infarction [33]. Chen et al. revealed that the p38 MAPK signaling pathway regulates the metalloproteinase ADAM17, promoting ACE2 detachment and impairing its cardioprotective function, thereby playing a crucial role in the pathogenesis of HF following myocardial infarction [44]. Additionally, Dawuti et al. found that salvianolic acid A alleviates cardiac inflammation, fibrosis, and diastolic dysfunction by inhibiting the p38 MAPK/CREB signaling pathway, thereby attenuating HF with preserved ejection fraction [34].

In summary, extensive studies have demonstrated that the MAPK signaling pathway plays a crucial role in the pathogenesis of HF through multiple pathophysiological mechanisms, particularly by influencing cardiac hypertrophy and myocardial fibrosis. Moreover, modulating the MAPK signaling pathway offers potential therapeutic strategies for regulating HF progression [28], presenting new avenues for drug development and innovative approaches to prevent, treat, and slow HF progression in clinical practice (Figure 1).

## 4. MAPK Signaling Pathway and Myocardial Ischemia/Reperfusion Injury

Myocardial ischemia–reperfusion injury (MIRI) is a pathological process associated with various diseases, leading to tissue necrosis and organ dysfunction. Ischemia refers to the restriction of blood supply, resulting in severe tissue hypoxia. The restoration of blood flow to ischemic tissue can paradoxically cause further damage, a phenomenon known as MIRI [45].

A growing body of research highlights the critical role of ferroptosis in the pathogenesis and progression of various cardiovascular diseases, including MIRI [46]. Ferroptosis is a distinct form of cell death characterized by iron-dependent lipid peroxidation [47]. Chen et al. demonstrated through in vivo and in vitro experiments that MIRI could be alleviated by inhibiting the MAPK signaling pathway and suppressing ferroptosis in cardiomyocytes [48]. Similarly, Liu et al. found that suppression of the Sat1 gene in a rat model alleviated MIRI by regulating ferroptosis via the MAPK/ERK pathway [49]. Collectively, these findings suggest that the MAPK signaling pathway plays a regulatory role in MIRI pathogenesis by modulating ferroptosis.

Beyond its role in ferroptosis, MAPK has also been implicated in other pathophysiological mechanisms, particularly apoptosis. As demonstrated by Wang et al., the process of MIRI has been shown to promote the expression of GADD45A [50]. This activation subsequently triggers the p38 MAPK/JNK pathway, which reduces cell viability and promotes apoptosis. Furthermore, the study showed that inhibiting ERK1/2, JNK, and p38 downregulates MIRI-induced EGR1 expression and reduces apoptosis in cardiac microvascular endothelial cells [50]. Lv et al. found that inhibiting PARP1 activation and p38-MAPK/JNK1/2 phosphorylation attenuated cardiac injury and apoptosis, thereby protecting cardiomyocytes [51]. These findings indicate that MAPK signaling activation contributes to cardiomyocyte apoptosis, thereby exacerbating MIRI. He et al. demonstrated that MAPK inhibition attenuated MIRI by suppressing oxidative stress and excessive autophagy in both in vivo and in vitro models [52].

Furthermore, in a mouse model, Jin et al. found that JNK activation upregulated mitochondrial fission factor (Mff), leading to lethal mitochondrial fission, excessive mitochondrial autophagy, and mitochondrial energetic disturbances, ultimately contributing to MIRI [53]. In a mouse model, Ahmad et al. identified GSK-3α as a key mediator of I/R-induced inflammation and cell death via MAPK activation. Consequently, this exacerbated cardiac injury and impaired post-I/R remodeling [54].

In conclusion, the findings above indicate that the MAPK signaling pathway plays a central role in MIRI pathogenesis, particularly through ferroptosis and apoptosis. Extensive research has demonstrated that inhibiting MAPK phosphorylation can mitigate MIRI, offering novel therapeutic approaches for its management (Figure 2).

## 5. Cardiac Hypertrophy and the Role of the MAPK Signaling Pathway

In the adult heart, the quantity of cardiomyocytes remains predominantly constant, with alterations in cell size occurring in response to external stimuli. Cardiac hypertrophy serves as an adaptive response to physiological and pathological stimuli, helping to maintain cardiac output under stress [55,56]. Hypertrophy is categorized into physiological and pathological types, each governed by distinct molecular pathways. Persistent hypertrophy can result in cardiac remodeling, functional decline, heart failure, and, in some cases, sudden cardiac death [57]. In certain cases, pathological hypertrophy may progress to heart failure [56]. The MAPK signaling system plays a critical role in regulating cardiomyocyte development, apoptosis, metabolism, and fibrosis, making it a key regulator of both physiological and pathological hypertrophy.

Research suggests that various components of the MAPK signaling cascade, including the ERK family and MAP2K3, play distinct roles in the progression of cardiac hypertrophy [55,58]. MAP2K3, an upstream regulator of p38 MAPK, is known to enhance TGF-β–dependent pro-fibrotic and hypertrophic gene expression. In hypertensive murine models, ATF3—a factor primarily expressed in cardiac fibroblasts under hypertensive conditions—is upregulated in hypertrophic cardiomyopathic hearts. In this model, upregulated ATF3 functions as a negative feedback regulator to confer cardiac protection. This mechanism inhibits hypertrophy by suppressing MAP2K3 and consequently downregulating p38/TGF-β signaling [58].

Beyond its role in pathological ventricular hypertrophy, evidence suggests that ERK, a key component of the MAPK pathway, contributes to cardiac hypertrophy by promoting cardiomyocyte regeneration. A study on neonatal mouse embryos demonstrated that ERK, as a downstream effector, participates in the BMP7-BMPR/ACVR signaling pathway, stimulating cardiomyocytes to enter the S phase and facilitating adult cardiomyocyte re-entry after myocardial injury. This process triggers cardiomyocyte cytoplasmic division and may enhance proliferation [59]. In vitro studies demonstrated that recombinant Agrin enhances cardiomyocyte proliferation in mouse and human-induced pluripotent stem cells through an ERK-mediated mechanism [22]. Collectively, these studies suggest that MAPK promotes hypertrophy by facilitating cardiac regeneration through ERK-dependent mechanisms.

Moreover, MAPK signaling has been linked to the modulation of autophagy in the heart. Studies suggest that ghrelin exerts cardioprotective effects by activating the MAPK/ERK pathway, which subsequently promotes mTOR signaling and suppresses excessive autophagy [60]. This may represent a promising strategy for mitigating the progression of cardiac hypertrophy. Conversely, another study demonstrated that allicin regulates autophagy through the MAPK/ERK/mTOR pathway in neonatal rat cardiomyocytes under Ang II-induced hypertrophic conditions, effectively suppressing cardiac hypertrophy [57]. Another in vitro study showed that the JNK/MAPK pathway, as a downstream signaling cascade, is regulated by *lincRNA-EPS* also to promote macrophage autophagy [61].

In conclusion, the MAPK signaling pathway plays a multifaceted regulatory role in cardiac hypertrophy, with specific branches involved in cardiomyocyte proliferation, metabolic regulation, fibrosis, and autophagy (Figure 3). p38 MAPK contributes to pathological hypertrophy and fibrosis through TGF-β–mediated gene expression, whereas ERK signaling is implicated in both pathological hypertrophy and cardiac regeneration under specific conditions. Furthermore, MAPK/ERK/JNK signaling regulates mTOR-dependent autophagy, thereby influencing the progression of hypertrophy.

## 6. The Role of MAPK in Atherosclerosis Formation and Progression

Atherosclerosis constitutes a worldwide cardiovascular issue [62]. This condition accounts for a significant proportion of global mortality and remains the leading cause of death worldwide. It is believed to result from a complex interplay of multiple factors [62,63]. The increasing global incidence of atherosclerosis underscores the urgency of understanding its underlying mechanisms.

Previous research has demonstrated that atherosclerosis progression involves multiple complex mechanisms, including monocyte adhesion, macrophage transformation and lipid uptake, foam cell formation, innate and adaptive immune responses, as well as processes such as adhesion, chemotaxis, differentiation, dedifferentiation, lipid uptake, angiogenesis, and apoptosis [63,64]. Atherosclerotic plaques consist of various components, including endothelial cells, lipids, and tissue fragments [65]. Among these complex mechanisms, the MAPK signaling pathway is recognized as a key regulator of foam cell production, endothelial cell function, and inflammatory responses.

A defining characteristic of atherosclerosis is the uptake of changed lipids by subendothelial macrophages, leading to the development of foam cells [66]. Foam cells primarily originate from the accumulation of free and esterified cholesterol within macrophages, though they may also form through the transformation of endothelial cells (ECs) and vascular smooth muscle cells (VSMCs) [66,67]. p38 MAPK overexpression in macrophage-enriched areas of atherosclerosis plaque has been detected in male rabbits using Western blot and immunohistochemistry techniques [68]. Its activation may be triggered by pro-inflammatory factors, extracellular stress, intracellular DNA damage, or protein misfolding [69]. Studies have identified p38 MAPK as a key regulator in the positive feedback loop of foam cell formation, as evidenced by its activation through oxLDL and eLDL, which enhances macrophage LDL uptake via PPARγ signaling [69]. Furthermore, both p38 MAPK and JNK signaling pathways have been shown to promote foam cell formation via monocyte-platelet aggregates (MPAs) in in vitro experiments using THP-1 cells [70]. Research suggests that inhibition of H1R signaling enhances the p38 MAPK-LIPG pathway, leading to increased foam cell formation in mouse bone marrow-derived macrophages (BMDMs) [71]. Moreover, JNK signaling, another major component of the MAPK pathway, is closely linked to macrophage inflammation and foam cell formation. A 2020 study demonstrated that ginsenoside compound K (CK) effectively inhibited the NF-κB, p38, and JNK MAPK pathways in an in vitro experiment using *RAW264.7* macrophages, significantly altering foam cell metabolism and thereby influencing atherosclerosis progression [72].

Various vascular cell types have been shown to play a role in atherosclerosis formation through MAPK signaling. Research utilizing apolipoprotein-deficient murine models demonstrated that physical activity could enhance lysine lactylation of Mecp2 (Mecp2k271la), hence suppressing the Ereg-induced MAPK signaling pathway. This mechanism partially inhibits the inflammatory response induced by oxLDL, thereby reducing EC adhesion to macrophages [73]. Endothelial dysfunction (ED) is a critical factor in vascular remodeling [74]. Under adverse conditions such as hypertension, hypercholesterolemia, diabetes, and smoking, ECs can undergo endothelium-mesenchymal transition (EndMT), a key process closely linked to atherosclerosis development [75,76]. The study found that TGF-β, as a molecule specific to chronic inflammation, is believed to be a key factor in determining EndMT. It can affect EndMT through SMAD signal transduction SMAD-independent pathways such as ERK, PI3K/Akt, and JNK/p38 [77]. Additionally, vascular smooth muscle cells (VSMCs) and VSMC-derived cells serve as the primary sources of plaque cells and extracellular matrix throughout different stages of atherosclerosis. VSMCs differentiate into various plaque cell phenotypes, including fibrocap extracellular matrix-producing cells, macrophage-like cells, foam cells, mesenchymal stem cell-like cells, and osteochondrogenic cells [65]. In a study on PG-LPS-induced signaling pathways involved in human aortic smooth muscle cell (HASMC) proliferation and migration, phosphorylation levels of p38 MAPK at Thr180/Tyr182 and ERK at Thr202/Tyr204 were assessed using Western blot analysis, with peak levels detected within five minutes of stimulation [78]. These findings suggest that p38 MAPK/ERK/JNK activation regulates vascular smooth muscle cell migration in response to LPS stimulation, thereby influencing atherosclerosis progression.

Beyond the above mechanisms, atherosclerosis is increasingly recognized as a chronic inflammatory-driven process [65]. Among the MAPK subtypes, p38 MAPK is crucial in inflammatory signaling [69]. A study on a high-fat diet-induced atherosclerotic mouse model analyzed the infiltration of vascular inflammatory proteins. This study found that downregulation of pathway proteins p38, ERK, and NF-κB inhibited inflammatory signaling, reducing pro-inflammatory proteins (IL-1α, IL-1β, IL-6) while increasing anti-inflammatory proteins (IL-10) [79]. Another biogenic analysis combined with the validation of atherosclerosis mouse models showed that among the influencing factors related to signal transduction, the TNF signaling pathway, HIF-1 signaling pathway, MAPK signaling pathway, NF-kappa B signaling pathway, VEGF signaling pathway, and PI3K-Akt signaling pathway occupy the main part. These pathways converge on a common target, such as MAPK1. This suggests that MAPK contributes to atherosclerosis progression through synergistic interactions with other signaling pathways. In the experiment on synchronous atherosclerosis mice, it was found that drug inhibition of the above pathway significantly reduced the expression of inflammatory factors in the serum of atherosclerosis mice. Levels of TNF-α, IL-6, IL-1β, and IL-18 were significantly reduced, reflecting reduced levels of inflammation in the body. Overall, this intervention suppressed the inflammatory response, improved blood lipid levels, and reduced plaque formation in atherosclerosis mice [80]. 

In conclusion, atherosclerosis, a prevalent cardiovascular disease on a global scale, is characterized by intricate molecular mechanisms that underpin its initiation and progression. The MAPK signaling pathway plays a pivotal role in regulating foam cell formation, endothelial cell function, vascular smooth muscle cell migration, and inflammatory responses. As outlined in this section, key branches of the MAPK pathway, including p38 MAPK, ERK, and JNK, contribute to atherosclerosis progression by modulating LDL uptake, macrophage-mediated inflammation, EndMT, and smooth muscle cell proliferation. Furthermore, the MAPK pathway engages in crosstalk with other critical signaling pathways, such as TNF, PI3K-Akt, and NF-κB, collectively orchestrating inflammatory responses and vascular remodeling in atherosclerosis (Figure 4).

## 7. Prospective Therapeutic Insights

Optimal arterial architecture and functionality are crucial for preserving the physiological integrity of the cardiovascular system. Vascular remodeling, defined as structural modifications in blood vessels, is intricately associated with numerous cardiovascular disorders. MAPK signaling serves as a vital epigenetic modulator of gene expression within the regulatory systems involved [77]. Considering its importance, targeting MAPK and its associated upstream or downstream molecules may yield a breakthrough in treatment approaches for vascular and ventricular remodeling. Recent pharmacological research has concentrated on MAPK and its associated pathways, identifying many chemicals and medicines that may prevent, stop, or delay the advancement of cardiovascular illnesses. These findings present encouraging avenues for the formulation of novel therapeutic regimens.

### 7.1. Chinese Medicines Targeted on MAPK Signaling Pathway

Traditional Chinese medicine (TCM) has gained increasing attention for its potential role in cardiovascular disease treatment, particularly through modulation of the MAPK signaling pathway. Many TCM formulations and bioactive compounds have been found to regulate key molecular mechanisms involved in cardiac hypertrophy, heart failure (HF), myocardial ischemia/reperfusion injury (MIRI), and atherosclerosis. By targeting specific branches of the MAPK cascade, these therapies can inhibit pathological remodeling, reduce oxidative stress, and modulate inflammatory responses.

Shexiang Tongxin Dropping Pill (STDP), a prevalent traditional Chinese medicine prescription for coronary artery disease, has recently been shown to ameliorate chronic HF in a murine model by blocking the ERK/MAPK pathway, thereby diminishing cardiomyocyte hypertrophy and myocardial fibrosis [36]. Danshensu A (Salvianolic Acid A) has been shown to mitigate HF with preserved ejection fraction (HFpEF) via modulating the p38 MAPK/CREB signaling pathway [34]. Salvianolic acid B has been shown to exert a significant anti-apoptotic effect in the context of myocardial ischemia/reperfusion injury. This effect is achieved through the attenuation of the JNK MAPK apoptosis signaling pathway, thereby modulating the process of ferroptosis and apoptosis [81]. Olive Picroside (OP), a naturally occurring strong antioxidant, has been shown to attenuate MIRI and improve adverse cardiac remodeling in vivo by inhibiting the MAPK pathway [52]. Bioinformatics analysis of Scutellariae Radix-Coptidis Rhizoma (QLYD) revealed 225 potential target genes, predominantly linked to oxidative stress responses, inflammatory regulation, epithelial cell apoptosis, and blood coagulation, with the MAPK signaling pathway recognized as a crucial regulatory network [80]. Furthermore, a 2024 investigation demonstrated that a traditional Chinese medicinal combination, called the Zexieyin formula, recorded in the Huangdi Neijing, has considerable anti-atherosclerotic properties. This compound uniquely modulates the MAPK/NF-κB pathway and selectively promotes macrophage polarization towards the anti-inflammatory M2 phenotype, hence effectively diminishing the risk of unfavorable cardiovascular events [79].

In summary, Chinese medicines targeting the MAPK signaling pathway offer promising therapeutic potential for cardiovascular diseases.

### 7.2. Western Medicines Targeting the MAPK Signaling Pathway

Western medicine has explored various pharmacological agents that regulate the MAPK signaling pathway to combat cardiovascular diseases. These therapies aim to inhibit ferroptosis and foam cell formation, reduce myocardial apoptosis, and alleviate atherosclerosis by targeting key molecular mechanisms. Recent research has identified several compounds with cardioprotective effects through MAPK pathway modulation, offering new avenues for therapeutic intervention.

Oxymatrine (OMT) has demonstrated the ability to inhibit the production of proteins associated with the MAPK pathway, potentially mitigating HF through this method [82]. Jin et al. discovered that Oroxylin A has the capacity to inhibit the process of ferroptosis, thus protecting myocardium and reducing MIRI, by modulating the MAPK-Nrf2 pathway [83]. In addition, Dapagliflozin has been demonstrated to attenuate myocardial I/R injury by reducing ferroptosis through the MAPK signaling pathway [48]. Lv et al. demonstrated that 25-hydroxycholesterol has the potential to induce cardioprotective effects in MIRI by means of inhibiting PARP1 activation and the phosphorylation of p38-MAPK and JNK1/2 against cardiomyocyte apoptosis [51]. A 2023 investigation showed that papain, a proteolytic enzyme, can inhibit MAPK signaling, thereby decreasing MPA-induced foam cell production and ultimately mitigating atherosclerosis progression [70]. Antioxidants like kaempferol have demonstrated anti-atherosclerotic properties by obstructing particular ion channels and calcium influx, hence indirectly influencing the NF-κB/MAPK pathway and suppressing foam cell formation. This method was subsequently corroborated in a 2024 investigation [84]. In addition, modulating endothelial cell activity through MAPK regulation has surfaced as a possible intervention strategy. One technique entails exogenous lactate delivery, which has been shown to inhibit Ereg expression in endothelial cells (ECs), resulting in the control of MAPK activity and a decrease in atherosclerosis progression [73].

In conclusion, Western medicines targeting the MAPK pathway present promising strategies for cardiovascular disease management, underscoring the importance of MAPK modulation in future drug development.

### 7.3. Challenges and Future Outlook in MAPK-Targeted Therapies

Emerging study discoveries offer vital insights into the application of MAPK inhibitors or analogous medications for the management of cardiovascular disorders; nonetheless, it is crucial to acknowledge that MAPK-related factors may demonstrate contradictory effects based on the particular situation. Distinguishing the diverse downstream consequences of MAPK signaling may enhance drug development tactics and expand therapeutic possibilities.

Research indicates that the dynamic phase separation of HIP-55 can inhibit excessive P38/MAPK activation in β-adrenergic receptor-mediated HF, therefore offering cardioprotective benefits. Phosphorylation-defective HIP-55 variants forfeit their protective role against HF, highlighting the necessity for meticulous evaluation in medication development [85].

Moreover, in the context of mitigating cardiovascular problems, it has been observed that PPARγ-activating medications utilized for the treatment of type II diabetes may elevate the chance of congestive HF. The activation of GPR40 by PPARγ stimulates ERK1/2, hence modulating downstream PPARγ responses. Should a biased GPR40 agonist be created to selectively activate p38 MAPK and PPARγ while circumventing ERK1/2 activation, this therapeutic strategy could possess wider clinical applicability [86].

### 7.4. Future Directions in MAPK-Targeted Cardiac Therapy

Future studies should elucidate the specific functions of MAPK signaling in cardiac hypertrophy and HF, emphasizing the contributions of various branches to fibrosis, inflammation, and metabolic dysfunction. The ERK-mediated equilibrium between heart regeneration and pathological remodeling is particularly noteworthy, along with the roles of p38 MAPK and JNK in fibrotic development and maladaptive hypertrophy. Comprehending these relationships may yield profound insights into the mechanisms propelling disease advancement.

Furthermore, accurate control of MAPK signaling offers a viable pathway for the development of tailored therapeutics. Identifying critical regulatory points within the MAPK cascade may enable future research to discover innovative therapeutic techniques that improve heart repair and reduce pathological remodeling. This may result in more efficient, individualized treatment strategies for HF, cardiomyopathy, and cardiac regeneration, ultimately enhancing clinical outcomes for impacted patients (Figure 5).

## 8. Conclusions

The MAPK signaling pathway has been demonstrated to modulate a number of processes that contribute to the progression of cardiovascular disease, including inflammation, fibrosis, apoptosis, autophagy, and endothelial dysfunction [19]. Research has indicated that the MAPK signaling pathway plays a regulatory role in heart failure, cardiac hypertrophy, atherosclerosis, and myocardial ischemia–reperfusion injury (Table 1).

A comprehensive review of recent studies shows that ERK, p38, and JNK are differentially implicated in cardiovascular disease pathogenesis. ERK is involved in heart regeneration, hypertrophy, and metabolic control, while p38 MAPK and JNK are involved in inflammation, fibrosis, and maladaptive remodeling [21,26,28]. In the context of HF, the MAPK signaling pathway serves as pivotal elements in the progression of the condition by exerting influence on cardiac hypertrophy and following myocardial fibrosis. In regard to MIRI, the MAPK signaling pathway exerts its influence on the process of HF by affecting ferroptosis and apoptosis in cardiomyocytes. Furthermore, in the context of cardiac hypertrophy, the MAPK signaling pathway fulfills a dual regulatory role in the pathogenesis of this condition, with certain branches of the pathway promoting cardiomyocyte proliferation, metabolic control, fibrosis, and autophagy. In atherosclerosis, the MAPK pathway modifies macrophages and regulates foam cell metabolism.

It is evident that the MAPK signaling pathway represents a pivotal pathogenic mechanism and a viable cardiovascular disease treatment target. The development of MAPK-targeted therapeutic interventions, encompassing studies on molecular processes, pharmacological interventions, and therapeutic applications, holds significant potential for the management of heart failure, atherosclerosis, myocardial damage, and related cardiac disorders. These findings allow reasonable speculation that the MAPK signaling pathway inhibition by anti-oxidative stress agents, or the inhibition of pathophysiological mechanisms downstream of the MAPK signaling pathway, are also potential therapeutic targets. Pharmacological inhibitors, traditional Chinese medicine formulations, and novel bioactive compounds that modulate MAPK activity may offer novel therapeutic approaches to prevent pathological remodeling and improve cardiovascular function.

Despite the fact that the role of MAPK in cardiovascular disorders has been the subject of increased research, there are still several hurdles to overcome. The context-dependent effects of MAPK signaling, in conjunction with its intricate interaction with other cellular pathways, pose a significant challenge in the design of highly tailored and effective medicines. It is recommended that future research endeavors focus on the identification of different MAPK, the development of targeted therapeutic interventions with minimal off-target effects, and the investigation of patient-specific signaling profiles as a basis for therapy regimens.

## Figures and Tables

**Figure 1 ijms-26-02667-f001:**
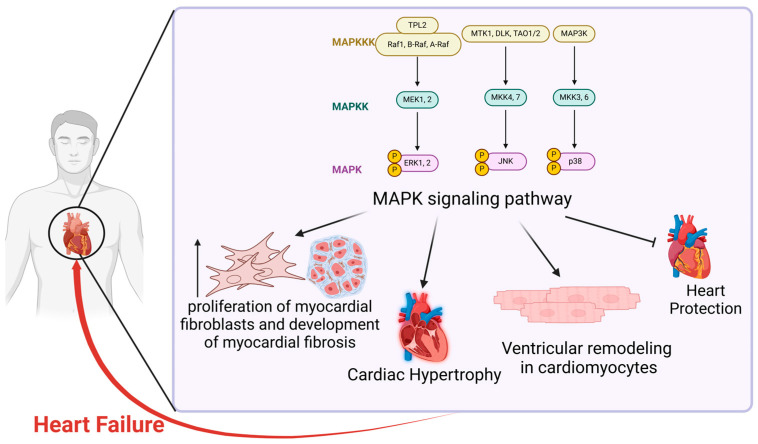
The role of the MAPK signaling pathway in heart failure. The figure illustrates the function of the MAPK signaling pathway in heart failure. Members of the MAPK family have the capacity to induce or exacerbate heart failure through a range of potential pathophysiological mechanisms, the most significant of which include the promotion of myocardial fibrosis and the induction of myocardial hypertrophy. In recent years, there has been an increasing recognition of the impact on ventricular remodeling in cardiomyocytes and the indirect hindrance of cardioprotective functions. While MAPK signaling pathways may be involved in the development of heart failure, they do not constitute the primary causative agents. The MAPK signaling pathway can promote the proliferation of myocardial fibroblasts and the development of myocardial fibrosis, which can ultimately result in heart failure. Additionally, the MAPK signaling pathway has been observed to enhance cardiac hypertrophy and ventricular remodeling in cardiomyocytes, which can also contribute to heart failure. However, it is important to note that the MAPK pathway can also inhibit the function of heart protection, thereby leading to heart failure.

**Figure 2 ijms-26-02667-f002:**
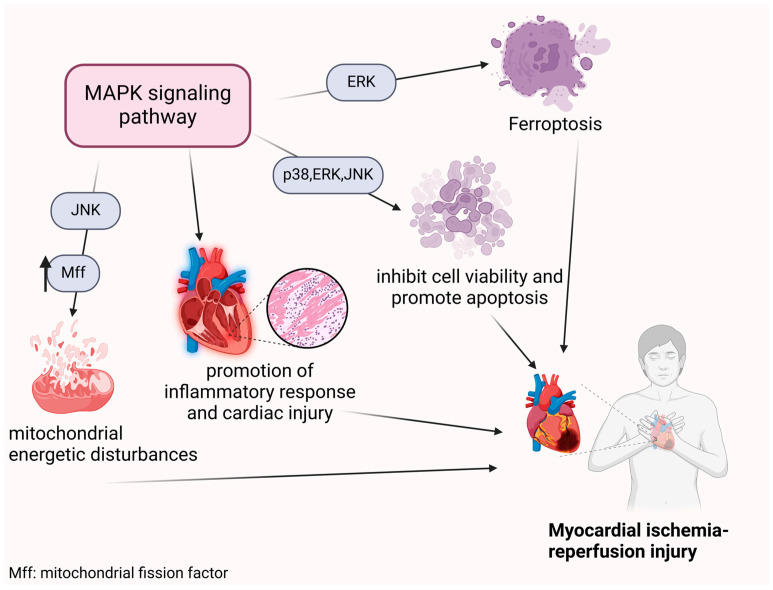
Effect of the MAPK signaling pathway on MIRI. The MAPK signaling pathway has been demonstrated to exacerbate MIRI through a range of pathophysiological mechanisms, the most common of which include the induction of ferroptosis and apoptosis in cardiomyocytes. Furthermore, the MAPK signaling pathway can directly or indirectly contribute to inflammatory responses resulting in cardiomyocyte injury. Activation of the MAPK signaling pathway also leads to upregulation of Mff, which in turn disrupts mitochondrial energetic disturbances and ultimately results in myocardial ischemia-reperfusion injury. Collectively, these pathophysiological mechanisms have the potential to exacerbate MIRI to a variable extent.

**Figure 3 ijms-26-02667-f003:**
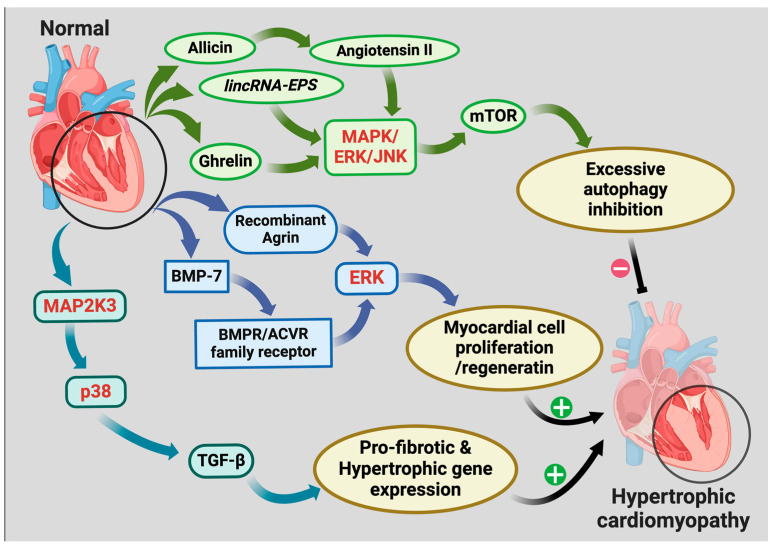
The Function of the MAPK Signaling Pathway in Cardiac Hypertrophy. This figure depicts the function of the MAPK signaling pathway in cardiac hypertrophy. MAP2K3 stimulates p38 MAPK, facilitating TGF-β-dependent pro-fibrotic and hypertrophic gene expression, whereas ATF3 modulates this process in cardiac fibroblasts. ERK, regulated by BMP-7 and recombinant agrin, facilitates myocardial cell proliferation and regeneration. The MAPK/ERK-mTOR axis regulates autophagy, and its dysfunction results in hypertrophic cardiomyopathy. This underscores the dual function of MAPK in pathological remodeling and cardiac healing. Different colors mean different signaling factors groups of different hypertrophic mechanisms. The colored arrows mean enhancement or stimulation, and the dark arrows with +/- mean promotion/inhibition.

**Figure 4 ijms-26-02667-f004:**
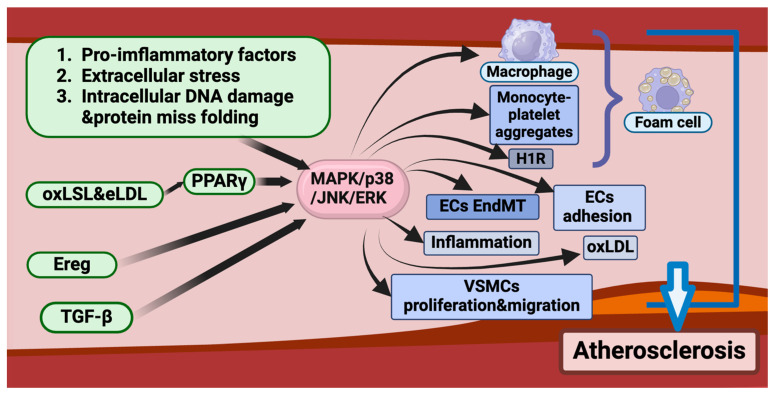
The role of the MAPK signaling pathway in atherosclerosis development and progression. This diagram illustrates the role of the MAPK signaling pathway in the formation and progression of atherosclerosis. Various stimuli, including pro-inflammatory factors, extracellular stress, intracellular DNA damage, and protein misfolding, can activate MAPK/p38/JNK/ERK signaling. This activation promotes multiple pathogenic processes, including macrophage transformation, monocyte-platelet aggregate formation, endothelial-to-mesenchymal transition (EndMT), endothelial cell adhesion, inflammation, and vascular smooth muscle cell (VSMC) proliferation and migration. Additionally, modified lipids such as oxLDL and eLDL enhance MAPK activation through PPARγ, leading to increased foam cell formation. Other regulatory factors, such as Ereg and TGF-β, further contribute to MAPK-mediated endothelial dysfunction and atherosclerosis progression. These interconnected mechanisms highlight MAPK as a critical regulator of vascular pathology, making it a potential therapeutic target for atherosclerosis treatment.

**Figure 5 ijms-26-02667-f005:**
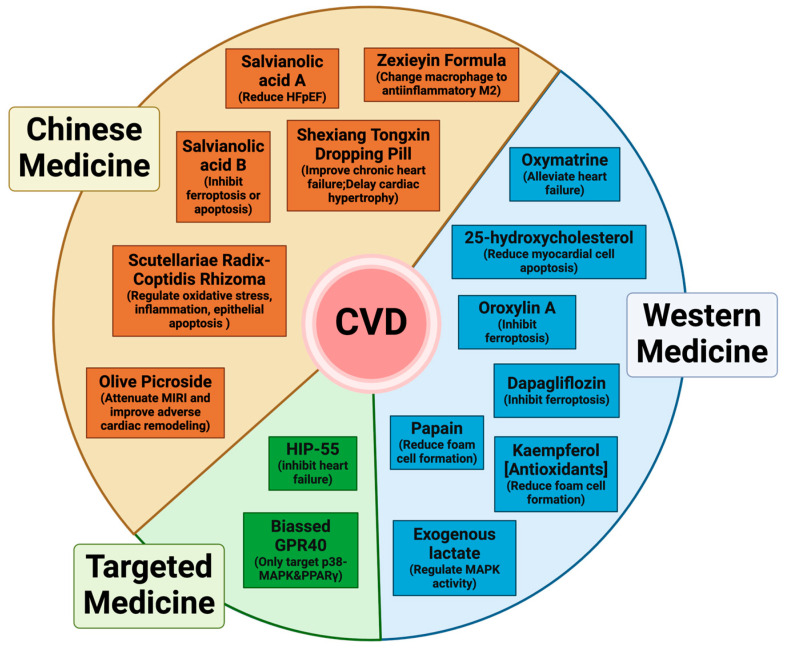
MAPK-targeted therapeutic strategies for cardiovascular diseases. This diagram illustrates different therapeutic strategies for cardiovascular diseases (CVDs), categorized into Chinese medicine, Western medicine, and targeted medicine. The Chinese medicine section includes Salvianolic acid A, Salvianolic acid B, Shexiang Tongxin Dropping Pill, and others, which improve cardiovascular health by regulating the MAPK signaling pathway, inhibiting apoptosis, or preventing ferroptosis. The Western medicine section features Oxymatrine, 25-hydroxycholesterol, Dapagliflozin, and more, primarily acting through MAPK pathway inhibition, reduction in myocardial cell apoptosis, and suppression of ferroptosis. The targeted medicine section includes HIP-55 and biased GPR40 agonists, which focus on precisely modulating the MAPK signaling pathway to explore more specific treatment approaches. These diverse therapeutic strategies highlight the critical role of the MAPK pathway in cardiovascular diseases and offer potential directions for personalized treatment in the future.

**Table 1 ijms-26-02667-t001:** Summary of the MAPK signaling pathway and their effects.

Category	In Vitro, In Vivo Models	Effect	Reference
Heart Failure	diabetes-related mice	cardiac damage and myocardial fibrosis	[38]
	GSK-3α-mediated mouse in vivo and in vitro	myocardial fibrosis	[39]
	chronic HF-related mice in vivo	myocardial fibrosis and cardiomyocyte hypertrophy	[36]
	in mouse model	pathological cardiac hypertrophy	[41]
	in vivo HF model	regulate mitochondrial homeostasis	[42]
	chronic HF-related rat in vivo	ventricular remodeling	[43]
	HFpEF mouse model	cardiac inflammation, fibrosis, and diastolic dysfunction	[34]
	animal model of post-MI HF and in vitro hypoxic cell model	activation of ADAM17	[44]
Myocardial Ischemia/Reperfusion Injury	MIRI rat model and hypoxia/reoxygenation (H/R)-induced H9C2 cardiomyocytes	ferroptosis in cardiomyocytes	[48]
	rats and H9C2 cells	ferroptosis	[49]
	I/R injury rat model and hypoxia/reoxygenation (H/R) cellular model	Inhibition of cell viability and promotion of apoptosis	[50]
	IR injury model mice	cardiac injury and apoptosis	[51]
	in vivo and in vitro experiments	oxidative stress and excessive autophagy	[52]
	in vivo and in vitro experiments	mitochondrial energetic disturbances	[53]
	in a mouse model and AC16 cardiomyocyte model	cardiac injury, inflammatory response, and effect on remodeling	[54]
Cardiac Hypertrophy	hypertension murine model in vivo and in vitro experiments	pathological hypertensive ventricular hypertrophy	[58]
	zebrafish and mouse in vitro and in vivo	heart regeneration and cardiomyocyte proliferation	[59]
	in vivo and in vitro experiments	cardiomyocyte regeneration and proliferation	[22]
	in H9c2 cells	excessive autophagy	[60]
	in vivo and in vitro experiments	autophagy	[57]
	THP-1 cells and atherosclerosis rat model	foam cell production and monocyte-platelet aggregate	[70]
Atherosclerosis	Apolipoprotein E-knockout mice	promotion of inflammation and foam cell formation	[71]
	RAW264.7 macrophages	macrophage inflammation and foam cell formation	[72]
	high-fat-diet-induced apolipoprotein-deficient mouse model and mouse aortic endothelial cells	promotion of the regression of atherosclerosis	[73]
	human aortic smooth muscle cells	proliferation and migration of human vascular smooth muscle cells	[78]
	high-fat-diet-induced vascular inflammation and vulnerable plaques in mice	increase in inflammation and reduction in plaque stability	[79]
	atherosclerosis mice model	inflammatory response, reduction of blood lipid levels, and improved plaque area	[80]

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
