# Peer review of "The Role of the MAPK Signaling Pathway in Cardiovascular Disease: Pathophysiological Mechanisms and Clinical Therapy"

_ijms, 2025, doi:10.3390/ijms26062667_

Round 1
Reviewer 1 Report
Comments and Suggestions for Authors
The paper “The role of MAPK signaling pathway in cardiovascular disease: pathophysiological mechanisms and clinical therapy” provides a comprehensive review of the possible mechanisms by which the very complex signaling network of mitogen activated protein kinases might be involved in the pathogenesis of some of the most frequent cardiovascular diseases, such as heart failure, cardiac hypertrophy, atherosclerosis etc. Although there are numerous reviews dealing with MAPK signaling and heart conditions, given the importance of the subject, new papers can still be valuable. After the introductory sections describing the complexity and significance of MAPK cascades, the authors examine the pathogenic role of these enzymes in specific diseases/conditions, and finish with a section dedicated to future therapeutic modalities.
The idea behind the review is valid, however there are substantial oscillations in the quality of the text. Some parts of the manuscript are more clear, some are imprecise and confusing (it is not always clear whether the humans or animals are in question, etc...). There is redundancy of statements, mainly regarding the importance of MAPK signaling. Similarly, there are also sentences that are too general or superficial, not adding much to the matter that is discussed in a specific segment of the text (for instance “Research indicates that the MAPK signaling pathway, characterized by its varied network, involves many components such as ERK1/2, JNK, and p38 MAPK, which interact with other pathogenic variables to influence different aspects of HF development”).
The final subsection “Prospective Therapeutic Insights” must be better structured-authors should either mention all the disorders with corresponding (potential) therapies, or avoid the subdivisions according to diseases, and give just some examples where therapeutic attempts have been made.
In other words, authors should carefully re-check the manuscript and focus their narrative onto actual mechanisms (fibrosis, ferroptosis, apoptosis...) connecting MAPK and cardiovascular disorders.
Lines 115-118: this entire section is unclear (what is post-cardiac damage, what is a structural pathway, and how they concur in sustaining homeostasis?).
Line 184: the sentence “Furthermore, numerous studies have demonstrated that the progression of HF can be regulated by modulating the MAPK signaling pathway...”, needs some references.
Line 215: delete “thereby contributing to the development of MIRI”, because you said it at the beginning of the sentence.
Lines 311-312: the sentence must be rephrased because it is unclear.
English language and style should be improved.
Comments on the Quality of English Language
Can be improved.
Reviewer 2 Report
Comments and Suggestions for Authors
This review aims to address the role and involvement of the MAPK pathway in cardiovascular disease in order to propose new clinical treatment strategies for CVD. The intentions are interesting, but the structure and development of the different chapters need to be reviewed.
1) It is known in the literature that mechanisms such as inflammation and oxidative stress underlie the pathogenesis of cardiovascular disease. It would be interesting to further investigate and write about the role and relationship of MAPKs, e.g. p38, with these mechanisms, as they are important in the pathological context under consideration.
2) The information provided in this article is important, but I suggest including a table or paragraph to report the data and results (e.g. from in vivo and in vitro experiments). There is extensive scientific literature on the modulation of MAPKs in the various contexts discussed in this review. This could improve the clarity of the article.
3) The paragraph entitled ‘The role of MAPKs in the formation and progression of atherosclerosis’ is not very well developed. I would suggest expanding it by analysing the relationship referred to in the title a little further. The text refers to the involvement of VSMCs or MAPK subtypes, I would suggest that these important aspects of plaque progression should be explored in more depth.
4) I suggest inserting a schematic illustration encompassing all the possible therapeutic approaches addressed in the paper, this would help the immediacy in conveying the intended message.
Round 2
Reviewer 1 Report
Comments and Suggestions for Authors
Authors have significantly improved their manuscript. It can be accepted for publication now.
Reviewer 2 Report
Comments and Suggestions for Authors
The work in the present form has improved significantly, so I have no further comments to make